# “Build Your Village”—Conducting the Village Test on Cognitively Impaired Patients: A First Journey into Alzheimerland

**DOI:** 10.3390/brainsci14060523

**Published:** 2024-05-21

**Authors:** Michelangelo Stanzani-Maserati, Maddalena De Matteis, Luca Bosco, Flavia Baccari, Corrado Zenesini, Micaela Mitolo, Chiara La Morgia, Roberto Gallassi, Sabina Capellari

**Affiliations:** 1IRCCS Istituto delle Scienze Neurologiche di Bologna, Via Altura 3, 40139 Bologna, Italy; maddalenadematteis@yahoo.it (M.D.M.); flavia.baccari@ausl.bologna.it (F.B.); corrado.zenesini@isnb.it (C.Z.); micaela.mitolo@unibo.it (M.M.); chiara.lamorgia@unibo.it (C.L.M.); sabina.capellari@unibo.it (S.C.); 2Istituto di Psicologia Individuale “Alfred Adler”, 10125 Torino, Italy; lucabosco@yahoo.it; 3Dipartimento di Scienze Biomediche e NeuroMotorie, Università di Bologna, 40126 Bologna, Italy; roberto.gallassi@unibo.it

**Keywords:** Alzheimer’s disease, dementia, Village Test, personality, projective techniques

## Abstract

Background: This work aimed to study the Village Test (VT) in a group of patients with Alzheimer’s disease (AD) and compare the results with those of a group of patients with mild cognitive impairment (MCI) and controls. Methods: A total of 50 patients with AD, 28 patients with MCI, and 38 controls were evaluated. All participants underwent the VT and an extensive neuropsychological evaluation. Results: The mean ages of the participants were 74.4 years for those with AD, 74 for those with MCI, and 70.2 for the controls. The AD group built smaller and essential villages with a scarce use of pieces, a poor use of dynamic pieces, and scarce use of human figures. All constructions were often concentrated in the center of the table. Conclusions: The villages built by the AD group represent a cognitive and affective coarctation and indicate a sense of existential disorientation and isolation. The VT is a useful aid for getting in touch with the inner emotional and existential states of patients with AD, and it could represent a complementary screening tool for orienting cognitive impairment diagnoses.

## 1. Introduction

Alzheimer’s disease (AD) is a neurodegenerative disorder of increasing prevalence in the Western world, being the most common cause of dementia. In the U.S., 10.8% of the population aged 65 and older has Alzheimer’s disease, and the percentage of affected people increases with age: 5.0% of people aged 65 to 74, 13.1% of people aged 75 to 84, and 33.3% of people aged 85 and older [1]. AD is a disease that affects not only cognition, but also affectivity and behavior, leading to a progressive deficit of individual and social functioning.

Diagnosing AD is a complex procedure that implies neuropsychological and neuroradiological exams. A neuropsychological assessment is made up of tests for the screening of patients for a detailed study of cognitive functions. Screening tests are useful in guiding neuropsychological evaluations, and they are usually easy to administer [2].

Recently, a complementary method to the traditional conception of screening cognitively impaired patients was proposed, based on projective techniques currently used in the study of personality, especially in those cases where a neuropsychological screening evaluation is not clearly unambiguous [3,4,5,6,7,8]. In fact, personality can be used not only to determine and condition patients’ behavioral disorders, but also to represent the background on which individual cognitive functioning has organized and developed. The alteration of the cognitive functioning of a patient with Alzheimer’s disease leaves a different trace on this background compared to in a subject without Alzheimer’s disease, which can be compared in terms of age and education and can be explored indirectly through projective techniques based on a projective psychic mechanism that uses unconscious mental dynamics.

Among personality tests, projective techniques, mainly those that involve non-verbal praxic functions (e.g., drawings, object manipulation, etc.), might be better suited than questionnaire tests for detecting such a functional trace in a way that does not directly involve cognition, especially verbal cognitive functions.

These latter types of tests are generally easier to perform with cognitively impaired patients, facilitating the unconscious expression of affection and personality. In this sense, we have previously described the use of two projective personality tests, such as the Tree Drawing Test (TDT) and the Human Figure Drawing Test (HFDT), to show the possibility of differentiating patients with AD from those with mild cognitive impairment (MCI) and healthy subjects [3,4,6].

Among projective personality tests of the non-verbal type, the Village Test (VT) is a typical test that mainly involves praxic abilities. First described by Henri Arthus in 1949, it can be easily administered to children and adults, as well as to individuals or groups of subjects [9,10,11,12,13]. Like the TDT and HFDT, the VT is a representation of one’s own self, and it can be used, in addition to within a psychodiagnostic battery, during psychotherapy as a ludic-expressive tool for children by creating a transitional space in which the therapist and the child playfully touch and move profound aspects of the self. With adults, the use of this instrument offers the possibility of observing a three-dimensional representation of one’s internal world. The psychotherapist can thus help the subject to move around in a village, to live there, to become acquainted with their own internal organization, to grasp the sense of the various places of the self, and to modify some parts of it, promoting self-reflection and therefore psychic reorganization [11,12,13].

Globally, the basic idea of the VT is that a game, as well as graphics in paper-and-pencil tests and the drawing of scribbles, allows something from the internal space to be transferred into a delimited external space [9,10]. These graphic expressions, playful constructions, and, in general, “projective creations” are the product of the encounter between the internal space of the subject and the external space [14,15]. The projection in a test or through a technique that allows objects to be assembled in space (e.g., the VT and Sceno test) essentially reflects the way in which the subject inhabits their body. If the subject inhabits a body, then the body inhabits a space, and the subject can organize it in their image and likeness. The construction of a village, therefore, reveals a representation of the subject’s body, both as a container and as a frontier between inside and outside, as well as the weaknesses of its organization and the fragility of the body ego [15].

Compared to other projective tests that expect two-dimensional stimuli and solicit a visual and verbal restructuring (e.g., Rorschach, TAT/CAT), the village has three-dimensionality and is made up of independent elements, thus also making it possible to observe the sensorimotor manipulation of the subject and their global motor skills.

The test involves building a village starting with a kit that includes a variable number of colored wooden pieces, 175 or 200 depending on the version used, depicting buildings (houses, shops, church, castle), elements of nature (people, animals, trees), and other more or less related pieces (means of transportation, fountains, cylinders, cubes, shingles, parallelepipeds). The subject decides how to interpret them and how to use them, without time limits. The delivery plan is to build a village on a table measuring 120 cm × 74 cm: “Here is what it takes to build a village or a city. Use whatever material you want”. At the end of the construction, there will be an inquiry of free-form questions to better understand what has been built in detail.

At the end of the test, it is thus possible to carry out a quantitative and a qualitative analysis of the configuration of the village (how the subject builds), of its symbolism (what the subject builds), and of its topography (where the subject builds) [9,12].

The quantitative analysis provides for the calculation of the total execution time and the construction start latency time; the measurement of the length and width of the village according to the main dimensional axes; the approximate size of the village area; the total number of pieces used; the type and number of the main groups of pieces according to the criterion of belonging to the group of living/non-living, human/non-human, dynamic/static.

The qualitative analysis of the configuration of the village concerns above all the way in which the subject approaches the test. This modality is similar to the one with which the subject approaches the tests in the external world, in particular those in which she/he has to start from a situation of emptiness and absence of definition or chaos (reflected in the test by the empty table and by the pile of randomly placed pieces) and to reach a situation of greater definition, coherence, and harmony, as in the case in which a problem is solved, an objective is achieved, or a greater degree of psychological and social well-being is reached. It can then be observed whether the subject builds in an impulsive, hasty, poorly reasoned way, if she/he shows anxiety, if she/he speaks or is silent, if she/he needs to ask questions, if she/he needs long periods of reflection, if she/he has frequent second thoughts and therefore disassembles and reassembles entire areas that have already been built, if she/he devalues or enhances the result of her/his work, if there is coherence between the construction and the description she/he gives, if she/he moves around the table and adopts multiple points of view or remains static or even blocked.

The analysis of the configuration of the village also concerns its location and dimensions in the space of the table. The location and dimensions of the village in space can reflect the way in which the subject inhabits her/his body, her/his experience of her/his own positioning in the world, as well as the extent of their range of action in the world [11,12]. People with a good degree of psychological and social well-being, as well as with good cognitive abilities, tend to use more pieces of the kit and more table space than those who do not have these characteristics. In situations of very strong relational closure, which lead to a poor investment in the outside world (as, for example, in depression and autism), people usually make very limited use of kit pieces and table space [11,12,13].

The final configuration that the subject gives to the village reflects the internal psychic organization. The border of the village, understood as a perimeter (the “container”) and as a frontier between inside and outside, gives indications on the characteristics of its borders, its somatopsychic structure, as well as its defenses and the type of relationship it establishes with the internal and external world [11,12,13]. In this sense, we observe how the subject has placed the buildings, built roads, squares, fences, or barriers, and how the village was shaped (closed/open, soft/rigid, square, circular, islands, radial), obtaining an impression: disintegration, chaos, harmony, vitality, emptiness, stasis, entrenchment, isolation, openness [11,12].

The analysis of the symbolism of the village mainly concerns its content, which also speaks of the inner world of the subject. What she/he has inserted and what she/he has omitted, the type of relationship that exists between the various sub-areas and the various characters, the closure or opening to the outside world, the internal dynamism, the relationship between solids and voids, the connotation that the subject gives to the various elements, taken together, help to obtain a picture of the characteristics of the subject’s self [11,12,13]. The placement or otherwise of people and animals, and the description that is made of them, give us an indication on the relational and affect-driven aspects that characterize the subject [13].

Finally, the topographical analysis considers which area of the table the village as a whole is located in or detects the characteristics of the subsets of the village (e.g., wood, farm, zoo, school, station, etc.) that fall within a specific area of the table to which we have given a connotation and which in turn reflects the existential, psychological, emotional, relational, and psychopathological characteristics of the subject [11,12,13].

In particular, the third of the table on the left represents the past, what we are derived from, our childhood, our family environment (parental area); the central third represents the present moment, the current psychological situation of the subject; the third on the right represents the future, what we are headed towards, the outside world, society (social area). The lower half recalls concrete, bodily, unconscious aspects. The upper half recalls abstract, symbolic aspects relating to thought. At the center of the table is the subject’s ego, i.e., the area around which one’s core identity can be assembled [9,10,11,12,13].

Furthermore, topological analysis should give insight into the dynamics with which the village was built, considering the way the progressive occupation of space on the table has taken place [10,11,12,13].

VT has been mainly studied in pediatric life and for psychological purposes. To our knowledge, there are no data available in the literature describing the characteristics of villages built by AD patients with different degrees of cognitive impairment. Since knowledge of the inner world of cognitively impaired patients is very important to better empathize and establish more suitable and personalized treatments, we evaluated VT in a group of AD patients and compared the results with those of a group of MCI patients and a group of controls.

## 2. Materials and Methods

We evaluated consecutive outpatients, 50 AD and 28 MCI, referred for cognitive disorders over a 1-year period by their relatives and physicians or who spontaneously presented themselves to the Cognitive Disorders Center of IRCCS Istituto delle Scienze Neurologiche of Bologna, Italy. Patients were compared with a group of 38 healthy controls.

### 2.1. Inclusion and Exclusion Criteria

Patient inclusion criteria were as follows: (a) major or minor neurocognitive disorder according to DSM-V criteria [16]; (b) diagnosis of AD and MCI based on the international criteria [17,18].

Patient exclusion criteria were as follows: (a) current or previous neurological, psychiatric (e.g., major depressive disorder), and systemic diseases; (b) alcoholism or other substance abuse; (c) use of neuroleptics or other antipsychotics, tricyclic or serotoninergic antidepressants, trazodone, and benzodiazepines, considering their possible negative effects on cognition; (d) history of diseases with a significant impact on visual acuity (e.g., severe glaucoma, progressive cone dystrophy, severe cataracts, or macular degeneration); (e) history of relevant trauma to the upper limbs, severe arthrosis, or rheumatic pathologies with arthritic expression that could mechanically interfere with motor skills of the upper limbs.

All AD patients were taking acetylcholinesterase inhibitors or memantine at therapeutic doses.

A group of healthy controls was selected. These subjects did not have any past or present neurological, psychiatric, or general diseases, alcoholism or other substance abuse, history of diseases with a significant impact on visual acuity or on manual skills. They were selected mainly among patients’ relatives, whereas caregivers were excluded considering a possible interference of anxiety and depression.

All participants gave consent to personal data processing for research purposes and the protocol was approved by the Local Ethical Committee (CE AVEC 17066, 21 September 2017). All participants gave their informed consent to the study according to the Declaration of Helsinki. If the patient was not cognitively able to be adequately informed, consent was given by their legal representative. Data were collected according to the General Data Protection Regulation (GDPR) (Regulation (EU) 2016/679-Directive (EU) 2016/680).

### 2.2. Procedures

All patients and controls underwent VT and an extensive battery of neuropsychological tests, standardized in the Italian population, exploring global cognition and specific cognitive functions (Table 1) [19,20,21,22,23,24,25,26,27,28,29,30,31]. VT and neuropsychological tests were administered to patients by an examiner blind to the patient’s diagnosis.

Village Test: All patients and controls were requested to build a village on a table measuring 120 cm × 74 cm. The village kit was made up of 200 colored wooden pieces that reproduce realistic components of a village. In particular, these included 1 castle, 2 towers, 1 church, 1 bell tower, 8 adult human figures (4 females, 4 males), 4 child human figures (2 females, 2 males), 12 adult animals, 5 baby animals, 20 deciduous trees, 20 coniferous trees, 58 shingles, 6 cars, 1 truck, 1 locomotive, 35 houses, 8 cuboids, 8 roofs, 1 cuboid with barred windows, 1 blue cube, 1 fountain, 5 bridges/porticos, 1 cylinder. The average dimensions of the pieces ranged from a minimum of 4 mm (e.g., the width of shingles) to a maximum of 100 mm (e.g., the height of the bell tower). Pieces were originally randomly placed on another surface near the table.

Instructions were as follows: “Here is what it takes to build a village or a city. Use whatever material you want”. No limits of time were given.

At the end of the construction, a short inquiry was conducted with the following questions: (1) “Where do you enter from and where do you leave the village from?”; (2) “Who do you take with you to the village?”; (3) “Are there any villages bordering yours?”.

A quantitative analysis and a qualitative analysis of the VT were performed for each patient and control.

The quantitative analysis included determining the latency time of construction of the village and the total time of execution; the total and specific number of pieces used; the categorization of the pieces into human figures, living/non-living, dynamic/static; and the village area. Living pieces included human figures (adults and children), animals (adults and puppies), trees (deciduous and coniferous); non-living pieces included castle, towers, church, bell tower, shingles, houses, parallelepipeds, roofs, bridges and arcades, cube, cylinder, cars, locomotive, a truck, and a fountain. Dynamic pieces included cars, locomotive, one truck, and a fountain, while static pieces included castle, towers, church, bell tower, shingles, houses, parallelepipeds, roofs, bridges and arcades, cube, and a cylinder. The area of the village was approximately calculated by multiplying the major and the minor axis of the village.

Qualitative analysis considered the configuration of the village (how the subject built), its symbolism (what the subject built), its topography (where the subject built), and the answers to the questions given during the inquiry, rendering a global impression of what had been built. Qualitative analyses of the patients’ villages were summarized for the AD group and compared with the other groups of subjects, MCI and controls, considering the main common and distinctive characteristics.

Neuropsychological assessment: We evaluated global cognition by Mini Mental State Examination (MMSE) [19] and Brief Mental Deterioration Battery (BMDB) [21,22]. BMDB is derived from Mental Deterioration Battery [23] by discriminant function analysis procedures, allowing the inclusion of the smallest tasks with the highest correct classification and with a “Final Result” (FR) allowing a classification for each subject with respect to the threshold value of zero with negative scores considered as pathological. Mental Deterioration Battery, from which the BBDM is derived, consists of eight verbal and visuospatial tasks, which are scored using a method of equivalent points. Pathological tasks are those in which the subject’s performance is below the lower limit of the tolerance interval of 95% for a confidence level of 95%.

We also investigated short- and long-term verbal and visuospatial memory, visual memory, simple and selective visual attention, executive functions, abstract and concrete thinking, phonemic and semantic verbal fluency, constructional praxis, and visuospatial and perceptual functions.

### 2.3. Statistical Analysis

Descriptive statistics (including a detailed description of the components of the VT) were calculated separately for AD, MCI, and healthy controls, presenting categorical variables as absolute (n) and relative frequencies (%) and continuous variables as mean with standard deviation (SD) and as median with interquartile range (IQR). Normal distribution of quantitative variables was investigated by using the Shapiro–Wilk test.

The Kruskal–Wallis test (or analysis of variance, when appropriate) for continuous variables and chi-square test for categorical ones (or Fisher’s exact test, when appropriate) were used to assess differences in sociodemographics, VT indicators, and neuropsychological tests between the three groups of interest. Tests on continuous variables were followed by Dunn’s post hoc comparison.

Due to the distribution of the variables, adjustment for confounding factors (age, sex, and education) was performed using quantile regression, with VT indicators as dependent variables and belonging group as independent variables (with controls as group reference). The coefficient (of the 50th percentile) represents the median difference between the groups. The 95% confidence interval (95% CI) of the coefficients was reported [32].

Furthermore, Spearman’s Rho correlations were performed to assess the strength and direction of the association between VT indicators and cognitive functions in the AD group. Correlation coefficients between 0.5 and 1 were considered strong, between 0.3 and 0.5 moderate, and less than 0.3 weak. *p*-values of the correlation analyses were adjusted by the Benjamini–Hochberg false discovery rate (FDR) multiple testing correction [33]; after the correction, a q-value < 0.10 was considered statistically significant.

Statistical significance was set at *p* < 0.05. Analyses were performed using Stata v. 16.1 and R v. 4.3.1 software.

## 3. Results

This study includes 50 patients diagnosed with AD (22 women, 44%), 28 MCI (12 women, 42.9%), and 38 controls (20 women, 52.6%). Mean age is 74.4 ± 9.5 years for AD, 74 ± 6.1 years for MCI, and 70.2 ± 8.4 years for controls; mean education is 10.1 ± 4.5 years for AD, 8.4 ± 3.6 years for MCI, and 11.5 ± 3.8 years for controls; mean disease duration is 3 ± 1 years for AD and 1.2 ± 0.5 years for MCI. Data are reported in Table 2.

### 3.1. VT Quantitative Analysis

VT indicator values are reported in Table 3. Compared with controls, the AD group shows a significantly lower value for almost all indicators: time of execution (in seconds) (median = 399 and IQR = 188; 549 vs. median = 513.5 and IQR = 368; 657), village area (1009.5 and 736; 1947 vs. 3350 and 2303; 4543), % used pieces out of total (12.5 and 6; 24 vs. 37 and 24; 47), human figures (0 and 0; 1 vs. 6 and 3; 10), living pieces (8 and 2; 20 vs. 36 and 26; 46), non-living pieces (4.5 and 8; 25 vs. 35 and 21; 50), living/non-living ratio (0.5 and 0.2; 1 vs. 1.1 and 0.7; 1.4), dynamic pieces (0 and 0; 1 vs. 2.5 and 1; 5), static pieces (21.5 and 12; 38 vs. 58.5 and 39; 76), dynamic/static ratio (0 and 0; 0 vs. 0.1 and 0; 0.1). The latency time (in seconds) shows higher values in the AD group (3 and 2; 5 vs. 2 and 1; 3).

Compared with MCI, the AD group shows a significantly lower value in most of the indicators, except for time of execution (median = 339 and IQR = 188; 549 vs. median = 312.5 and IQR = 209; 477.5), latency time (3 and 2; 5 vs. 3 and 1.5; 7), and living/non-living ratio (0.5 and 0.2; 1 vs. 0.7 and 0.5; 1.1).

Compared with controls, the MCI group shows a significantly lower value for almost all the indicators, except for latency time (3 and 1.5; 7 vs. 2 and 1; 3).

After adjusting for age, sex, and education, results are confirmed. In general, the differences with the control group are more pronounced for the AD group than the MCI group. All coefficients and relative 95% CI are reported in Table 4.

### 3.2. VT Qualitative Analysis

Villages of AD patients are homogeneous and are built with a very limited use of the available material (Figure 1). They are built with little dynamism, using a small part of the available space and grouping the material mainly in the central or lower central area of the table. The pieces used tend to be grouped in a non-realistic way and with a poor overview to the point of representing villages that are almost destructured or reduced to a minimal nucleus and without a particular morphology. The most frequently used pieces, such as houses and trees, are at times placed incorrectly, for example, upside down. The use of the so-called strong pieces, such as the castle or the church, is reduced to a minimum; the use of composite pieces, such as the bell tower or the castle towers, is reduced, and if used, they are often used in an incomplete way, i.e., not combined with the pieces of which they are a part, such as the castle and the towers or the church and the bell tower. Living pieces are sparsely used and limited to animals and trees; the use of human figures is very scarce. Dynamic pieces are also underused. The most frequent access route to the village is from below or directly from the center, while the main exit route is from below. For about half of the patients, there are no similar villages near their own. In most cases, AD patients would bring their immediate family members with them into the village, especially their spouse, whether living or deceased. In general, the overall impression of these villages is of emptiness and lack of content.

Villages of MCI patients and controls are generally homogeneous. Compared to AD patients, MCI patients and controls use much more material and build villages with greater dynamism, occupying all or a good part of the available space, and with a greater formal definition. They use strong pieces and combine them in the most appropriate way; they insert animated material, especially human and animal figures, and dynamic material. Access to and exit from the village is variable and equally represented (from below, from above, from the center). For about 60% of patients and controls, there are similar villages near their own. In most cases, MCI patients and controls would mostly bring their children and less frequently their spouses into the village. In general, compared to AD patients, the overall impression of these villages is of more vitality and dynamism with the presence of relationships within the village and with the outside.

Although the villages built by MCI patients are qualitatively much more homogeneous to those of controls than of AD patients, some differences with controls are appreciable. Specifically, MCI patients use fewer pieces and build villages by occupying less of the available space on the periphery. Even if they combine pieces in an appropriate way, they include less animated material, especially the human figures.

### 3.3. Neuropsychological Assessment

Descriptive statistics of the neuropsychological tests, exploring global cognition and different cognitive functions, are reported in Table 5. Furthermore, only for the AD group, Spearman’s Rho correlations between neuropsychological tests and VT indicators were calculated. The main findings concern the association between village area and MMSEc (rho = 0.388; *p* = 0.012; *p*-adj = 0.271); human figures and MMSEc (0.339; 0.030; 0.227), Stroop time (0.338; 0.031; 0.227), and semantic fluency (0.337; 0.031; 0.227); living pieces and Rey–Osterrieth complex figure test delayed recall (0.386; 0.013; 0.114), phonemic fluency (0.376; 0.016; 0.114), and semantic fluency (0.351; 0.025; 0.135); dynamic pieces and MMSEc (0.331; 0.034; 0.224), copy design (0.378; 0.015; 0.224), judgment of line orientation (0.321; 0.041; 0.224), and Street’s completion test (0.347; 0.026; 0.224). All associations found were “moderate” or “weak”.

## 4. Discussion

VT is a projective technique used to assess personality by building a village from a kit of a variable number of colored wooden pieces following open-ended instructions. VT is mainly used in younger subjects together with other techniques, but it can also be used in adults and the elderly.

Personality, as the background of every behavior, includes cognition, and cognition is involved in the test, since praxic, semantic, visuospatial abilities, psychomotor activities, as well as global cognitive functioning are implied in manipulating objects and carrying out the global construction starting from single elements with different meaning. Thus, cognitively impaired patients could build villages with different characteristics compared to normal subjects.

In our study, we essentially show that AD patients build different villages compared to MCI patients and normal subjects. Specifically, their villages are smaller and essential with scarce use of pieces, mainly limited to houses and trees, with poor use of dynamic pieces like cars and trucks, and even more scarce use of human figures, being that the presence of living pieces is limited to trees and a few animals. All constructions are often concentrated in the center of the table with an empty space around.

From a psychological perspective, AD patients build villages without a specific morphology, limiting themselves to what is essential, thus showing little interest in details and expressing little need to satisfy themselves or others with elaborate constructions. Villages are built with little dynamism, using a small part of the available space and grouping the material mainly in the central or lower central area of the table. The other parts of the table are little used or not used at all. Due to the lack of dynamism in the construction phase and the concentration of a few pieces in the central area, it is thus not possible in these cases to carry out a topological analysis, i.e., an analysis of the spaces that are progressively occupied during the construction process.

In general, the villages built by AD patients have contiguity and proximity as the unifying criterion of the self; the poorness of the construction and the few pieces grouped with more or less care, close to each other, seem to indicate this clinging to a basic nuclear identity. There is therefore no disintegration, as in other clinical pictures (for example, in psychotic patients), but a progressive coarctation that leads the villages to be monads in the middle of the void. The global psychic sense is therefore the loss of contact with one’s self and the sense of disorientation.

In the construction phase, AD patients use few pieces, often in a central position, as if the field of consciousness were restricted to the central nucleus of the ego. In fact, the central positioning of the construction concerns in most cases that area of the space that at a symbolic level is considered the “area of the Ego” and the limitation to this area confirms the impoverishment of relational capacities, a narrowing of the self around a basic nucleus, and an identity progressively compromised by cognitive impairment.

In particular, the frequent use of houses and trees, sometimes positioned incorrectly, the scarce use of strong points (castle and church) and composite pieces (bell tower and towers), sometimes combined in an incongruous way, confirm the scarce ability to integrate the parts into a coherent and morphologically defined whole. The very scarce presence of human figures and the scarce use of living pieces, limited to trees and a few animals, as well as the paucity of dynamic pieces, with the great void around them, can indicate the loss of contact with others, the loss of references to personal experiences, their depressive experience, isolation, and existential disorientation. Overall, affective and relational skills seem to be scarcely activated. For less than half of AD patients, in fact, there are no similar villages near their own, indicating a progressive loss of contact with the social reality that surrounds them.

Conversely, the villages built by MCI patients and control subjects are homogeneous with each other and differ from those built by AD patients. MCI patients and controls use much more material; their construction tendency is to expand over most of the table and to create a more defined morphology. They insert animated material (human and animal figures) and dynamic material (means of transport); they think more than AD patients in the presence of other villages bordering theirs. Observing the villages, there is therefore a feeling of greater vitality and dynamism, the affective aspects are detectable, and there may be relationships within the village and with the outside.

Although the villages built by MCI patients are in general much more homogeneous to those of controls than of AD patients, it should be noted that MCI patients build villages that are a little smaller than those of normal subjects by occupying less of the available space on the periphery and use fewer of the available pieces by including less animated material, especially the human figures. Furthermore, MCI patients have both slightly longer latency times of construction of the village and shorter times of execution compared to controls, similar to AD patients. Taken together, these data could indicate that in MCI patients, the cognitive and psychoemotional changes occurring along the course of cognitive impairment could be represented by the changes that occur in village construction.

Like other projective techniques, the VT is easy to administer, allowing patients to express self-image with relatively little resistance, and thus it could be useful to study cognitively impaired patients, especially AD ones. In our AD patient group, in fact, probably due to a non-severe degree of cognitive impairment, all subjects were able to understand the instructions and carry out the task according to their ability.

In the last few years, it has been shown that other simple projective techniques, like the Tree Drawing Test (TDT), the Human Figure Drawing Test (HFDT), and The House Drawing Test, can be used in an attempt to discriminate patients suffering from AD from those with MCI and healthy subjects [3,4,6,7,8]. In particular, trees drawn by AD patients are significantly smaller compared to those drawn by MCI and healthy subjects, and they are poorly detailed, with a smaller crown compared to the trunk, while MCI patients draw trees intermediate in size between AD patients and healthy subjects. Similarly, AD patients draw smaller human figures than healthy subjects, with the body height being the part of the figure that is shorter compared to the head, with significantly fewer details, especially those related to the sensory organs and body details. Therefore, in AD patients, quantitative results on the TDT are akin to those on the HFDT, since the essential element of the drawing (trunk in TDT and head in HFDT) does not significantly change, if comparing AD and controls, as much as the crown and body that become relatively shorter. The growth of the crown and of the body, respectively, represents the expansion of self-identity and the psychological consolidation of future prospects along the adult life; therefore, the regressive trend of these variables in AD patients could be read as a sort of progressive coarctation of cognitive and emotional life.

Analogous data are also found in the drawing of houses of cognitively impaired patients [7]. In particular, houses drawn by such patients are simple, often small, with little or no doors or chimneys, revealing greater psychological and emotional inhibition compared to healthy subjects, with difficult access to imagination and a feeling of insecurity.

As we have seen, this global and progressive cognitive and affective coarctation is the same that we find represented in the qualitative and quantitative characteristics of the villages built by our AD patient sample.

We have also found that the characteristics of the villages built by AD patients indicate a loss of contact with oneself, a sense of existential disorientation and isolation, and an impoverishment of relational skills. These results are in line with what we had previously observed in relation to the analysis of the emotional state of AD patients through the use of the Lüscher color test, a projective technique that explores the inner emotional state through an unconscious selection of color preferences by means of standardized tables [5]. This simple tool, which requires minimal cognitive involvement, especially of language, shows differences in the emotional state of AD patients if compared to normal subjects. Globally, Lüscher color test data on such patients show that AD patients live with a feeling of personal change due to instability and emotional insecurity, experiencing physical discomfort and a bodily need to be welcomed in a favorable environment. AD is thus an experience of physical fragility and insecurity characterized by a significant need for a favorable environment and care. Similarly, these characteristics are also found in the analysis of the villages of our AD patient sample, confirming the VT as a technique suitable for the study of such patients.

Compared to our previous works related to the study of the TDT and the HFDT in AD patients, in this case, our patient sample was studied by means of an extensive battery of neuropsychological tests exploring specific cognitive functions and global cognitive functioning, instead of an analysis limited only to the assessment of global cognition through the MMSE [3,4,6]. Neuropsychological assessment clearly distinguishes AD and MCI patient groups from normal subjects, although the difference between MCI patients and controls is smaller, especially in constructional praxis. Thus, even villages built by MCI patients have more in common with controls than with AD patients, even if, as we have seen, some specific characteristics may already be more typical of the AD group.

The AD patient correlation analysis shows a positive association between the use of human figures, dynamic pieces, and a general cognition index such as the MMSE, as well as village area. Furthermore, the use of living pieces or human figures correlates with tests that explore visuospatial memory and language, while the use of dynamic pieces correlates with tests that explore constructional praxis and visuospatial functions. The greater use of living pieces, especially human figures, and dynamic pieces in VT is in general an expression of a greater mental complexity. Thus, the construction characteristics of the villages of AD patients could likely reflect the progressive cognitive deficit secondary to the course of the disease and the reduced use of living pieces, especially human figures, which also distinguishes MCI patients from controls, could probably be a sort of transition indicator between normality and the onset of cognitive deterioration.

Some limitations of this study should be noted. First, a larger sample of patients to enhance the generalizability of the findings is desirable as well as a larger group of healthy subjects, also considering possible genetic and environmental factors that could introduce selection bias. Second, a contextual comparison could be made with other projective techniques already studied, such as the TDT, the HFDT, and the House Drawing Test, or with more complex tests such as the Rorschach Test. Finally, further follow-up studies are needed to better study any changes in the VT over time, along with the progression of the disease, to better understand the possible influence of psychological and neurological determinants on repeated evaluations and to better explore the neuroanatomical association of the VT changes.

## 5. Conclusions

In conclusion, VT is an easy test to administer and, like the other projective techniques of the TDT, the HFDT, and the House Drawing Test, is suitable for a general characterization of cognitively impaired patients and the study of their psychoemotional aspects. It could thus be of aid in exploring the emotional experiences of such patients, getting in touch with their inner emotional and existential state, following them, and letting them make better-informed treatment choices. It could also represent a complementary screening tool for orienting cognitive impairment diagnosis.

## Figures and Tables

**Figure 1 brainsci-14-00523-f001:**
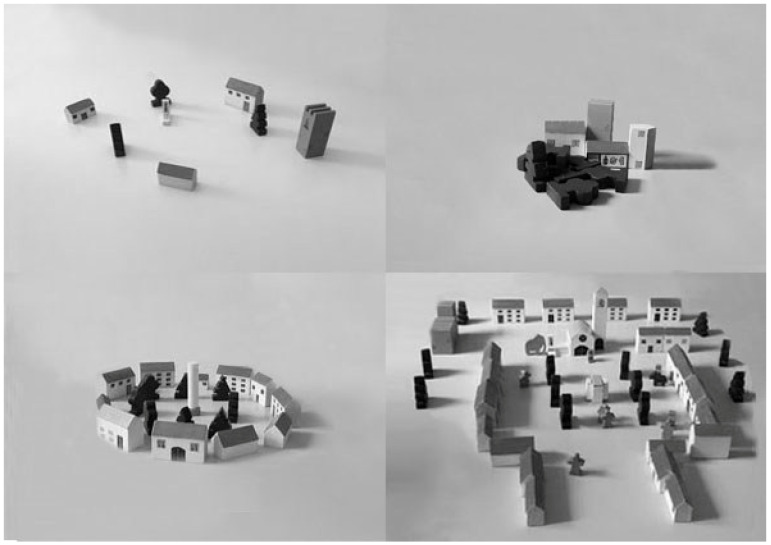
Examples of villages built by AD ((**upper left**) and (**right**)), MCI (**lower left**) patients, and controls (**lower right**).

**Table 1 brainsci-14-00523-t001:** Neuropsychological tests.

Cognitive Functions	Test
Global cognitive function indices	MMSE (range: 0–30) [19], MMSE corrected for age and education (range: 0–30; cut-off: <23.8) [20]
	Brief Mental Deterioration Battery (BBDM) with Final Result (FR) (cut-off: <0) [21,22]
Memory	Rey’s 15 Words *: immediate recall (range: 0–75; cut-off: <28.53); delayed recall * (range: 0–15; cut-off: <4.69) [23]
	Immediate Visual Memory * (range: 0–22; cut-off: <13.85) [23]
	Digit Span Forward (range: 0–9; cut-off: <3.75) [24]
	Corsi Block-tapping Test (range: 0-infinite; cut-off: <3.75) [25]
	Rey–Osterrieth Complex Figure Test: delayed recall (range: 0–36; cut-off: <6.20) [27]
Attention/Executive functions	Barrage Test * (time cut-off: ≥90; score cut-off: ≤9; errors cut-off: ≥2; result cut-off: >2.5) [21,22] Digit Span Backward (range: 0–9; cut-off: <2.65) [28]
	Stroop Test (time cut-off: >27.5; errors cut-off: >7.5) [26]
Abstract/concrete thinking	Analogies * (range: 0–20; cut-off: <13.92) [29]
Language	Verbal Fluency: phonemic (range: 0–infinite; cut-off: <17.35) [23]; semantic (range: 0–infinite; cut-off: <25) [30]
Constructional praxis	Copy Design: simple (range: 0–12; cut-off: <7.18) [23]
	Rey–Osterrieth Complex Figure Test: direct copy (range: 0–36; cut-off: <28) [27]
Visuospatial and perceptual functions	Judgment of Line Orientation Test (range: 0–30; cut-off: <19) [31] Street’s Completion Test (range: 0–14; cut-off: <2.25) [25]

* Tests included in the BBDM.

**Table 2 brainsci-14-00523-t002:** Demographic data of patients and controls.

		AD *n* = 50	MCI *n* = 28	Controls *n* = 38	*p*-Value	*p*-Value AD vs. MCI	*p*-Value AD vs. Ctrl	*p*-Value MCI vs. Ctrl
Age years	Mean (SD) Median [IQR]	74.4 (9.5) 76 [70; 82]	74 (6.1) 75.5 [72; 77.5]	70.2 (8.4) 70 [63; 74]	0.022	0.387	0.004	0.022
Sex f	N (%)	22 (44)	12 (42.9)	20 (52.6)	0.654	0.922	0.422	0.432
Education years	Mean (SD) Median [IQR]	10.1 (4.5) 8 [5; 13]	8.4 (3.6) 8 [5; 12.5]	11.5 (3.8) 13 [8; 13]	0.013	0.050	0.042	0.001

**Table 3 brainsci-14-00523-t003:** Village Test parameters.

	AD *n* = 50	MCI *n* = 28	Controls *n* = 38	*p*-Value	*p*-Value AD vs. MCI	*p*-Value AD vs. Ctrl	*p*-Value MCI vs. Ctrl
Time of execution (s) Median [IQR]	339 [188; 549]	312.5 [209; 477.5]	513.5 [368; 657]	0.007	0.422	0.003	0.005
Latency time (s) Median [IQR]	3 [2; 5]	3 [1.5; 7]	2 [1; 3]	0.273	0.424	0.077	0.079
Village area Median [IQR]	1009.5 [736; 1947]	1705.5 [885; 2964]	3350 [2303; 4543]	<0.001	0.025	<0.001	<0.001
% used pieces/total Median [IQR]	12.5 [6; 24]	19 [12.5; 33]	37 [24; 47]	<0.001	0.018	<0.001	<0.001
Human figures Median [IQR]	0 [0; 1]	1.5 [0; 8]	6 [3; 10]	<0.001	0.007	<0.001	0.002
Living pieces Median [IQR]	8 [2; 20]	16 [9.5; 30]	36 [26; 46]	<0.001	0.008	<0.001	<0.001
Non-living pieces Median [IQR]	14.5 [8; 25]	21 [15.5; 43]	35 [21; 50]	<0.001	0.028	<0.001	0.029
Living/non-living ratio Median [IQR]	0.5 [0.2; 1]	0.7 [0.5; 1.1]	1.1 [0.7; 1.4]	<0.001	0.115	<0.001	0.005
Dynamic pieces Median [IQR]	0 [0; 1]	1 [0; 4]	2.5 [1; 5]	<0.001	0.014	<0.001	0.034
Static pieces Median [IQR]	21.5 [12; 38]	32.5 [22.5; 59]	58.5 [39; 76]	<0.001	0.019	<0.001	0.004
Dynamic/static ratio Median [IQR]	0 [0; 0]	0 [0; 0.1]	0.1 [0; 0.1]	0.019	0.018	<0.001	0.229

**Table 4 brainsci-14-00523-t004:** Multivariable quantile regression models with VT indicators as dependent variables and diagnosis as independent variables (with controls as group reference) adjusted by sex, age, and education.

	Coeff	95% CI
Time of execution		
AD	−155.13	−234.33; −109.25
MCI	−181.63	−310.6; −60.85
Latency time		
AD	0.93	0.11; 1.41
MCI	0.89	0.32; 1.56
Village area		
AD	−2142.68	−2522.46; −1101.65
MCI	−1314.76	−2054.6; −488.97
% used pieces/total		
AD	−22.9	−28.98; −15.62
MCI	−16.26	−18.93; −6.52
Human figures		
AD	−4.89	−7.86; −3.88
MCI	−3.97	−7.74; −1.36
Living pieces		
AD	−23.86	−29.25; −19.6
MCI	−15.76	−20.27; −8.43
Non-living pieces		
AD	−20.2	−31.35; −11.78
MCI	−13.6	−24.17; −3.1
Living/non-living ratio		
AD	−0.68	−0.78; −0.33
MCI	−0.45	−0.62; −0.10
Dynamic pieces		
AD	−2	−3.5; −1.13
MCI	−1	−2.6; 0.43
Static pieces		
AD	−37.26	−41.01; −25.85
MCI	−23.36	−31.14; −10.01
Dynamic/static ratio		
AD	−0.1	−0.1; −0.01

**Table 5 brainsci-14-00523-t005:** Neuropsychological test results.

	AD	MCI	Controls	*p*-Value	*p*-Value	*p*-Value	*p*-Value
	*n* = 50	*n* = 28	*n* = 38		AD vs. MCI	AD vs. Ctrl	MCI vs. Ctrl
MMSEc							
Mean (SD)	18.7 (5.2)	24.5 (3.5)	27.9 (1.4)	<0.001	<0.001	<0.001	<0.001
Median [IQR]	19 [16; 22.7]	25.4 [22.5; 27.6]	28.1 [27.1; 29]				
BBDM Final Result							
Mean (SD)	−0.7 (1.3)	1.1 (0.9)	2.4 (0.5)	<0.001	<0.001	<0.001	<0.001
Median [IQR]	−1 [−1.4; 0.4]	1.2 [0.8; 1.6]	2.4 [1.9; 2.7]				
Rey’s 15 Words imm_c							
Mean (SD)	21.6 (5.6)	33.6 (6.9)	43.3 (6.7)	<0.001	<0.001	<0.001	<0.001
Median [IQR]	21.2 [17.1; 26.2]	33 [30.4; 37.6]	44.2 [36.4; 49.1]				
Rey’s 15 Words del_c							
Mean (SD)	2.4 (1.4)	5.6 (2.5)	8.8 (2.1)	<0.001	<0.001	<0.001	<0.001
Median [IQR]	2.8 [1.6; 3.3]	5.5 [4; 7.2]	8.3 [7.3; 10.4]				
Imm Visual Mem_c							
Mean (SD)	15.3 (4.5)	19 (2.5)	20 (1.9)	<0.001	0.001	<0.001	0.074
Median [IQR]	14.9 [12.3; 18.9]	19.2 [17.7; 20.5]	20 [19.5; 21.4]				
Digit Span Forward_c							
Mean (SD)	4.9 (1.2)	5.4 (1)	6.3 (1)	<0.001	0.066	<0.001	0.001
Median [IQR]	5.1 [4.1; 5.8]	5.6 [4.8; 6.3]	6.2 [5.7; 6.8]				
Corsi Block-tapping_c							
Mean (SD)	4.2 (4.3)	4.8 (0.8)	5.3 (1.2)	<0.001	<0.001	<0.001	0.109
Median [IQR]	3.6 [2.9; 4.2]	4.7 [4.3; 5.4]	5.1 [4.4; 6]				
Rey Fig delayed rec_c							
Mean (SD)	7 (4)	13.8 (8.2)	20 (5.8)	<0.001	<0.001	<0.001	<0.001
Median [IQR]	6.6 [4.5; 9.5]	11.6 [7.9; 19.8]	19.2 [16.3; 21.8]				
Barrage time							
Mean (SD)	109.7 (50.6)	64 (33)	46.5 (10.6)	<0.001	<0.001	<0.001	0.006
Median [IQR]	100 [65; 164]	58 [43; 66.5]	45 [38; 54]				
Digit Span Back_c							
Mean (SD)	3.1 (1.2)	3.9 (1.3)	5.5 (6)	<0.001	0.021	<0.001	0.005
Median [IQR]	3.1 [2.7; 3.6]	3.7 [3; 4.6]	4.6 [3.7; 5.3]				
Stroop time_c							
Mean (SD)	47.9 (51.3)	60.5 (148.5)	15.9 (7.9)	<0.001	0.499	<0.001	<0.001
Median [IQR]	38.4 [14.3; 56.3]	36.1 [19.8; 46.9]	15.3 [9.8; 21.3]				
Stroop err_c							
Mean (SD)	14.4 (9.9)	2.1 (5.2)	−1 (2.2)	<0.001	<0.001	<0.001	0.034
Median [IQR]	14.4 [4.6; 23.8]	0.6 [−1.7; 3.2]	−1 [−1.5; −0.5]				
Analogies_c							
Mean (SD)	9.4 (5.1)	14.1 (4.4)	18.2 (1.7)	<0.001	0.001	<0.001	<0.001
Median [IQR]	9.8 [5.7; 13.1]	14.8 [12.2; 17.4]	18.1 [16.7; 19.1]				
Phonemic fluency_c							
Mean (SD)	15.7 (8.8)	26.2 (10.9)	34 (7.8)	<0.001	<0.001	<0.001	0.001
Median [IQR]	16 [11.2; 22.4]	24.9 [16.2; 33.4]	32.8 [28.3; 40.3]				
Semantic fluency_c							
Mean (SD)	20.1 (6.8)	31.7 (11.2)	43.2 (6.2)	<0.001	<0.001	<0.001	<0.001
Median [IQR]	21 [17; 24]	32 [25; 38.5]	42.5 [39; 47]				
Copy Design_c							
Mean (SD)	9.3 (2.4)	11.1 (1.8)	11.6 (1)	<0.001	0.001	<0.001	0.116
Median [IQR]	9.8 [7.9; 11.1]	11.6 [10.1; 12.3]	11.7 [11.1; 12.4]				
Rey Fig copy_c							
Mean (SD)	21.5 (11.3)	32.1 (5.9)	36.5 (1.7)	<0.001	0.001	<0.001	0.001
Median [IQR]	24 [10.9; 31.5]	33.7 [31.4; 36]	36.8 [36; 37.3]				
Judg of Line Orient_c							
Mean (SD)	16 (8.8)	22.7 (4.2)	27.5 (3.7)	<0.001	0.007	<0.001	<0.001
Median [IQR]	18 [6; 23]	23.5 [19.5; 25.5]	28 [25; 30]				
Street’s Completion_c							
Mean (SD)	4.1 (2)	5.6 (2)	7.3 (1.7)	<0.001	0.003	<0.001	0.002
Median [IQR]	4 [2.8; 5.3]	5.8 [4.6; 6.9]	7.3 [6.5; 8.5]				

## Data Availability

The data are not publicly available due to patient privacy.

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
