# Peer review of "“Build Your Village”—Conducting the Village Test on Cognitively Impaired Patients: A First Journey into Alzheimerland"

_brainsci, 2024, doi:10.3390/brainsci14060523_

Round 1

Reviewer 1 Report

Comments and Suggestions for Authors

Summary

The manuscript presents a study utilizing the Village Test (VT) to explore spatial and projective abilities in patients with Alzheimer’s disease (AD), Mild Cognitive Impairment (MCI), and healthy controls. The authors aim to demonstrate the utility of VT as a non-verbal projective tool for distinguishing between different levels of cognitive impairment. The topic is highly relevant, given the increasing aging population. The use of a non-verbal, projective method like the VT is innovative and could potentially offer insights distinct from traditional neuropsychological assessments. The thoughtful analysis and clear presentation of methods and results contribute significantly to its field, with recommendations for future research paving the way for further explorations.

General concept comments:

*About half of the introduction section is the same as [1]. This is called "self-plagiarism" or "text recycling".

*The study might benefit from a larger and more diverse sample size to enhance the generalizability of the findings.

*The primary focus of the Village Test is to explore emotional experiences, conflicts, and unconscious processes. It is more about understanding an individual’s internal world rather than assessing cognitive functions such as memory, problem-solving, logical reasoning, or verbal ability. The manuscript reveals quantitative differences in villages built by different groups, but how to prove these differences are rooted in cognitive ability rather than the different psychological states of the groups? After all, it is reasonable to believe AD/MCI patients have different psychological states than healthy people.

*The authors stress that the VT does not directly imply verbal cognitive functions (line 59). However, do verbal cognitive functions affect how participants understand the instruction and task? If yes, then it will also affect the village built and should be discussed.

*The Village Test lacks standardization as each session is unique, and the outcomes depend greatly on individual circumstances and the organizer's perspective. Discussing whether longitudinal follow-up is planned or how repeated measurements might influence the understanding of VT outcomes in cognitive impairment progression could provide a fuller picture of the VT's utility over time.

Specific comments:

*Line 171 indicates no previous literature applying the Village Test on AD patients. To strengthen the background, the introduction should reference existing studies on the Village Test or similar projective tests used on other cognitively impaired patients. If no such studies exist, this should be highlighted as a gap in the literature, providing a stronger justification for the study.

*Line 198 indicates that the control group was primarily recruited from relatives of the patients. This selection method might introduce bias, as these healthy individuals could share environmental or genetic factors with the AD and MCI groups. Discussing any measures to account for potential confounding variables would strengthen the study's validity.

[1] Stanzani Maserati M, Matacena M, Baccari F, Zenesini C, Gallassi R, Capellari S, Matacena C. The Tree Drawing Test in Evolution: An Explorative Longitudinal Study in Alzheimer's Disease. Am J Alzheimers Dis Other Demen. 2022 Jan-Dec;37:15333175221129381. doi: 10.1177/15333175221129381. PMID: 36317413; PMCID: PMC10581108.

Author Response

Dear Reviewer,

Thank you very much for your observations and advice. I tried to follow your indications as follow.

General concept comments:

*About half of the introduction section is the same as [1]. This is called "self-plagiarism" or "text recycling".

I have rewritten and renovated the central part of the Introduction, being more synthetic but trying to keep the same concepts (from “Recently, a complementary…” till “…a way that does not directly imply cognition, especially verbal cognitive functions”).

*The study might benefit from a larger and more diverse sample size to enhance the generalizability of the findings.

I specified and declared this concept in the final part of the Discussion in the section dedicated to the limits of the work (“Some limitations of this study should be noted. First, a larger sample of patients to enhance the generalizability of the findings is desirable as well as a larger group of healthy subjects, also considering possible genetic and environmental factors that could introduce selection bias.”).

*The primary focus of the Village Test is to explore emotional experiences, conflicts, and unconscious processes. It is more about understanding an individual’s internal world rather than assessing cognitive functions such as memory, problem-solving, logical reasoning, or verbal ability. The manuscript reveals quantitative differences in villages built by different groups, but how to prove these differences are rooted in cognitive ability rather than the different psychological states of the groups? After all, it is reasonable to believe AD/MCI patients have different psychological states than healthy people.

Thanks for this observation of general concept. In this sense, I reorganized: 1) the final part of the Abstract, reversing the sequence of concepts, first relating to the psychoemotive aspects then relating to the possible complementary use of the test for cognitive screening; 2) The final part of the conclusions in the Discussion, according to the same conception.

*The authors stress that the VT does not directly imply verbal cognitive functions (line 59). However, do verbal cognitive functions affect how participants understand the instruction and task? If yes, then it will also affect the village built and should be discussed.

In the Introduction I attenuated this concept making it more a hypothesis than a determination (“Among personality tests, projective techniques, mainly those that imply non-verbal praxic functions (e.g. drawings, object manipulation etc.), might be better suited than questionnaire tests to detect such a trace through a way that does not directly imply cognition, especially verbal cognitive functions.”). 

In the Discussion I specified: “In our AD patients group, in fact, probably due to a non-severe degree of cognitive impairment, all subjects were able to understand the instructions given and carry out the task according to their ability.”

*The Village Test lacks standardization as each session is unique, and the outcomes depend greatly on individual circumstances and the organizer's perspective. Discussing whether longitudinal follow-up is planned or how repeated measurements might influence the understanding of VT outcomes in cognitive impairment progression could provide a fuller picture of the VT's utility over time.

I specified and declared this concept in the final part of the discussion in the section dedicated to the limits of the work (“Some limitations of this study should be noted. First, a larger sample of patients to enhance the generalizability of the findings is desirable as well as a larger group of healthy subjects, also considering possible genetic and environmental factors that could introduce selection bias. Second, a contextual comparison could be made with other projective techniques already studied, such as the TDT, the HFDT and the House Drawing Test, or with more complex tests such as the Rorschach Test. Finally, further follow-up studies are needed to better study changes in the VT over time along with the progression of the disease, to better understand the possible influence of psychological and neurological determinants on repeated evaluations and to better explore the neuroanatomical association of the VT changes.”).

Specific comments:

*Line 171 indicates no previous literature applying the Village Test on AD patients. To strengthen the background, the introduction should reference existing studies on the Village Test or similar projective tests used on other cognitively impaired patients. If no such studies exist, this should be highlighted as a gap in the literature, providing a stronger justification for the study.

Since specific scientific literature in this regard is not available, trying to follow what indicated by you, I added in the final part of the Introduction a phrase that tries to give greater strength to the reason why we developed our work of analysis of the villages built by cognitively impaired patients (“Since knowledge of the inner world of cognitively impaired patients is very important to better empathize and establish more suitable and personalized treatments, we therefore evaluated VT in a group of AD patients and compared the results with those of a group of MCI patients and a group of controls.”). 

*Line 198 indicates that the control group was primarily recruited from relatives of the patients. This selection method might introduce bias, as these healthy individuals could share environmental or genetic factors with the AD and MCI groups. Discussing any measures to account for potential confounding variables would strengthen the study's validity.

Thank you for your observation. I specified and declared this concept in the final part of the Discussion in the section dedicated to the limits of the work (“First, a larger sample of patients to enhance the generalizability of the findings is desirable as well as a larger group of healthy subjects, also considering possible genetic and environmental factors that could introduce selection bias.”).

Reviewer 2 Report

Comments and Suggestions for Authors

In the article entitled ""Build your village". The Test of the Village in cognitively impaired patients: a first journey into Alzheimerland", the authors show a study on the construction of a village that would represent a cognitive and affective impairment in patients with Alzheimer's disease, which would represent existential disorientation and isolation.

The study is very interesting and is a good approach to quantify cognitive impairment.

Some questions.

1. Of the patients with Alzheimer's disease, how much time has elapsed since they were diagnosed with the disease?

2. Did all Alzheimer's patients understand the instructions?

3. Were the patients with MCI under any treatment for the disease?

4.         Figure 1 (upper photos) shows the construction of the village by Alzheimer's patients. In general, did all patients make similar constructions?

5.         In table 3, the execution time and latency time in which units are they?

The study is very interesting, and the patients could be followed up in redoing the Village Test and perhaps it would be possible to teach and train them to build a typical Village so that they could follow as an example.

Author Response

Dear Reviewer,

Thank you very much for your questions and advice. I tried to answer your questions and adding information where possible.

Some questions.

1. Of the patients with Alzheimer's disease, how much time has elapsed since they were diagnosed with the disease?

In the Results section I added mean disease duration for the AD patients group and MCI (“mean disease duration is 3 ± 1 years for AD, 1.2 ± 0.5 years for MCI.”).

2. Did all Alzheimer's patients understand the instructions?

In the Discussion I specified: “In our AD patients group, in fact, probably due to a non-severe degree of cognitive impairment, all subjects were able to understand the instructions given and carry out the task according to their ability.”

3. Were the patients with MCI under any treatment for the disease?

MCI patients were not yet under a specific pharmacological treatment.

4. Figure 1 (upper photos) shows the construction of the village by Alzheimer's patients. In general, did all patients make similar constructions?

Yes, AD patients made similar constructions even if some details were different but the general impression was the same. 

5. In table 3, the execution time and latency time in which units are they?

Thank you for your observation. Latency time and execution time were in seconds. I specified it in the VT quantitative analysis section of Results and in the Table 3.